# TRANSFUSION: PREDICT THE NEXT TOKEN AND DIFFUSE IMAGES WITH ONE MULTI-MODAL MODEL

**Chunting Zhou**[μ*]  **Lili Yu**[μ*]  **Arun Babu**[μ]  **Kushal Tirumala**[μ]
**Michihiro Yasunaga**[μ]  **Leonid Shamis**[μ]  **Jacob Kahn**[μ]  **Xuezhe Ma**[σ]
**Luke Zettlemoyer**[μ]  **Omer Levy**[μ]

[μ] Work done at Meta
[σ] University of Southern California

## ABSTRACT

We introduce Transfusion, a recipe for training a multi-modal model over discrete and continuous data. Transfusion combines the language modeling loss function (next token prediction) with diffusion to train a single transformer over mixed-modality sequences. We pretrain multiple Transfusion models up to 7B parameters from scratch on a mixture of text and image data, establishing scaling laws with respect to a variety of uni- and cross-modal benchmarks. Our experiments show that Transfusion scales significantly better than quantizing images and training a language model over discrete image tokens. By introducing modality-specific encoding and decoding layers, we can further improve the performance of Transfusion models, and even compress each image to just 16 patches. We further demonstrate that scaling our Transfusion recipe to 7B parameters and 2T multi-modal tokens produces a model that can generate images and text on a par with similar scale diffusion models and language models, reaping the benefits of both worlds.

## 1  INTRODUCTION

Multi-modal generative models need to be able to perceive, process, and produce both discrete elements (such as text or code) and continuous elements (e.g. image, audio, and video data). While language models trained on the next token prediction objective dominate discrete modalities (OpenAI et al., 2024; Dubey et al., 2024), diffusion models (Ho et al., 2020; Rombach et al., 2022a) and their generalizations (Lipman et al., 2022) are the state of the art for generating continuous modalities (Dai et al., 2023; Esser et al., 2024b; Bar-Tal et al., 2024). Many efforts have been made to combine these approaches, including extending a language model to use a diffusion model as a tool, either explicitly (Liu et al., 2023) or by grafting a pretrained diffusion model onto the language model (Dong et al., 2023; Koh et al., 2024). Alternatively, one can quantize the continuous modalities (Van Den Oord et al., 2017) and train a standard language model over discrete tokens (Ramesh et al., 2021; Yu et al., 2022; 2023), simplifying the model's architecture at the cost of losing information. In this work, we show it is possible to fully integrate both modalities, with no information loss, by training a single model to both predict discrete text tokens and diffuse continuous images.

We introduce **Transfusion**, a recipe for training a model that can seamlessly generate discrete and continuous modalities. We demonstrate Transfusion by pretraining a transformer model on 50% text and 50% image data using a different objective for each modality: next token prediction for text and diffusion for images. The model is exposed to both modalities and loss functions at each training step. Standard embedding layers convert text tokens to vectors, while patchification layers represent each image as a sequence of patch vectors. We apply causal attention for text tokens and bidirectional attention for image patches. For inference, we introduce a decoding algorithm that combines the standard practices of text generation from language models and image generation from diffusion models. Figure 1 illustrates Transfusion.

In a controlled comparison with Chameleon's discretization approach (Chameleon Team, 2024), we show that Transfusion models scale better in every combination of modalities. In text-to-image

---

[*]Equal contribution.

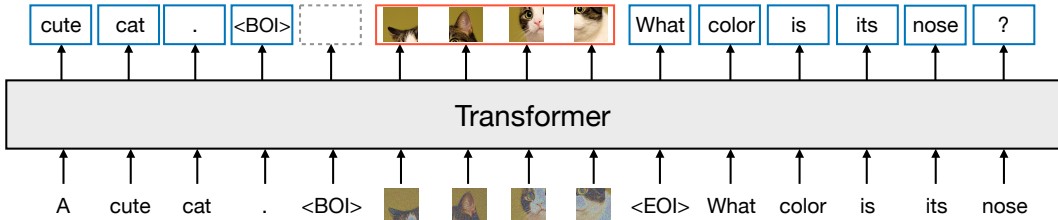

Figure 1: A high-level illustration of Transfusion. A single transformer perceives, processes, and produces data of every modality. Discrete (text) tokens are processed autoregressively and trained on the next token prediction objective. Continuous (image) vectors are processed together in parallel and trained on the diffusion objective. Marker BOI and EOI tokens separate the modalities.

generation, we find that Transfusion exceeds the Chameleon approach at less than a third of the compute, as measured by both FID and CLIP scores. When controlling for FLOPs, Transfusion achieves approximately $2\times$ lower FID scores than Chameleon models. We observe a similar trend in image-to-text generation, where Transfusion matches Chameleon at 21.8% of the FLOPs. Surprisingly, Transfusion is also more efficient at learning text-to-text prediction, achieving perplexity parity on text tasks around 50% to 60% of Chameleon's FLOPs.

Ablation experiments reveal critical components and potential improvements for Transfusion. We observe that the intra-image bidirectional attention is important, and that replacing it with causal attention hurts text-to-image generation. We also find that adding U-Net down and up blocks to encode and decode images enables Transfusion to compress larger image patches with relatively small loss to performance, potentially decreasing the serving costs by up to $64\times$.

Finally, we demonstrate that Transfusion can generate images at similar quality to other diffusion models. We train from scratch a 7B transformer enhanced with U-Net down/up layers (0.27B parameters) over 2T tokens: 1T text tokens, and approximately 5 epochs of 692M images and their captions, amounting to another 1T patches/tokens. Figure 7 shows some generated images sampled from the model. On the GenEval (Ghosh et al., 2023) benchmark, our model outperforms other popular models such as DALL-E 2 and SDXL; unlike those image generation models, it can generate text, reaching the same level of performance as Llama 1 on text benchmarks. Our experiments thus show that Transfusion is a promising approach for training truly multi-modal models.

## 2 BACKGROUND

Transfusion is a single model trained with two objectives: language modeling and diffusion. Each of these objectives represents the state of the art in discrete and continuous data modeling, respectively. This section briefly defines these objectives, as well as background on latent image representations.

### 2.1 LANGUAGE MODELING

Given a sequence of discrete tokens $y = y_1, ..., y_n$ from a closed vocabulary $V$, a language model predicts the probability of the sequence $P(y)$. Standard language models decompose $P(y)$ into a product of conditional probabilities $\prod_{i=1}^{n} P_\theta(y_i|y_{<i})$. This creates an autoregressive classification task, where the probability distribution of each token $y_i$ is predicted conditioned on the prefix of a sequence $y_{<i}$ using a single distribution $P_\theta$ parameterized by $\theta$. The model can be optimized by minimizing the cross-entropy between $P_\theta$ and the empirical distribution of the data, yielding the standard next-token prediction objective, colloquially referred to as *LM loss*:

$$\mathcal{L}_{\text{LM}} = \mathbb{E}_y \Big[ -\frac{1}{n} \sum_{i=1}^{n} \log P_\theta(y_i|y_{<i}) \Big] \tag{1}$$

Once trained, language models can also be used to generate text by sampling token by token from the model distribution $P_\theta$, typically using temperature and top-p truncation.

### 2.2 DIFFUSION

Denoising diffusion probabilistic models (a.k.a. *DDPM* or *diffusion models*) operate on the principle of learning to reverse a gradual noise-addition process (Ho et al., 2020). Unlike language models that typically work with discrete tokens ($y$), diffusion models operate over continuous vectors ($\mathbf{x}$), making

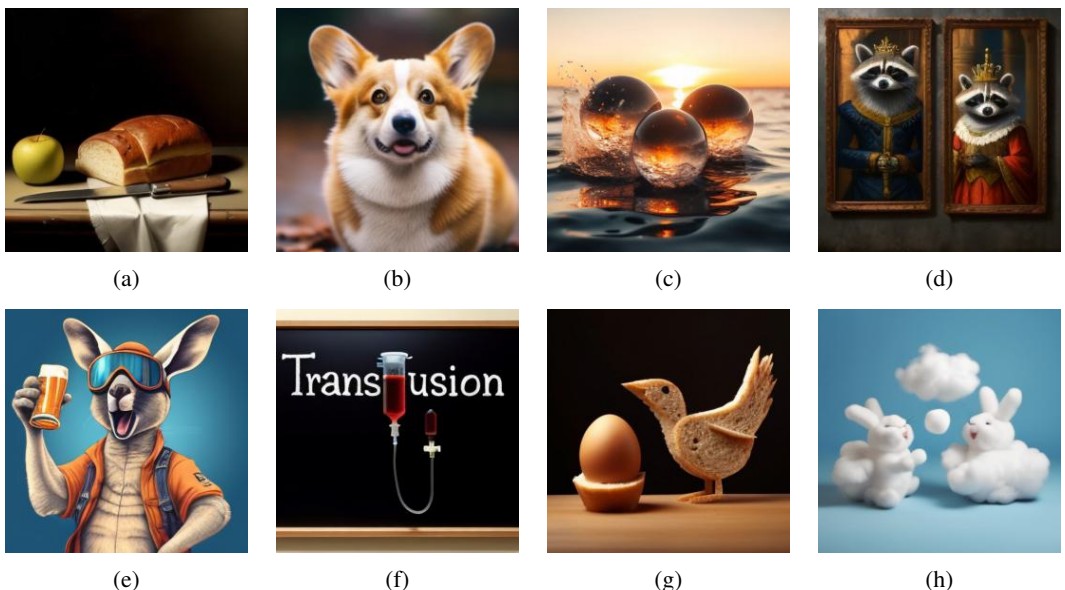

Figure 2: Generated images from a 7B Transfusion trained on 2T multi-modal tokens. Captions are: (a) A bread, an apple, and a knife on a table. (b) A corgi. (c) Three spheres made of glass falling into ocean. Water is splashing. Sun is setting. (d) A wall in a royal castle. There are two paintings on the wall. The one on the left a detailed oil painting of the royal raccoon king. The one on the right a detailed oil painting of the royal raccoon queen. (e) A kangaroo holding a beer, wearing ski goggles and passionately singing silly songs. (f) "Transfusion" is written on the blackboard. (g) an egg and a bird made of wheat bread. (h) A cloud in the shape of two bunnies playing with a ball. The ball is made of clouds too.

them particularly suited for tasks involving continuous data like images. The diffusion framework involves two processes: a forward process that describes how the original data is turned into noise, and a reverse process of denoising that the model learns to perform.

**Forward Process**    From a mathematical perspective, the forward process defines how the noised data (which serves as the model input) is created. Given a data point $\mathbf{x}_0$, Ho et al. (2020) define a Markov chain that gradually adds Gaussian noise over $T$ steps, creating a sequence of increasingly noisy versions $\mathbf{x}_1, \mathbf{x}_2, ..., \mathbf{x}_T$. Each step of this process is defined by $q(\mathbf{x}_t|\mathbf{x}_{t-1}) = \mathcal{N}(\mathbf{x}_t; \sqrt{1 - \beta_t}\mathbf{x}_{t-1}, \beta_t\mathbf{I})$, where $\beta_t$ increases over time according to a predefined noise schedule (see below). This process can be reparameterized in a way that allows us to directly sample $\mathbf{x}_t$ from $\mathbf{x}_0$ using a single sample of Gaussian noise $\boldsymbol{\epsilon} \sim \mathcal{N}(\mathbf{0}, \mathbf{I})$:

$$\mathbf{x}_t = \sqrt{\bar{\alpha}_t}\mathbf{x}_0 + \sqrt{1 - \bar{\alpha}_t}\boldsymbol{\epsilon} \tag{2}$$

Here, $\bar{\alpha}_t = \prod_{s=1}^{t}(1 - \beta_s)$, providing a useful abstraction over the original Markov chain. In fact, both the training objective and the noise scheduler are eventually expressed (and implemented) in these terms.

**Reverse Process**    The diffusion model is trained to perform the reverse process $p_\theta(\mathbf{x}_{t-1}|\mathbf{x}_t)$, learning to denoise the data step by step. There are several ways to do so; in this work, we follow the approach of Ho et al. (2020) and model the Gaussian noise $\boldsymbol{\epsilon}$ in Equation 2 as a proxy for the cumulative noise at step $t$. Specifically, a model $\boldsymbol{\epsilon}_\theta(\cdot)$ with parameters $\theta$ is trained to estimate the noise $\boldsymbol{\epsilon}$ given the noised data $\mathbf{x}_t$ and timestep $t$. In practice, the model often conditions on additional contextual information $c$, such as a caption when generating an image. The parameters of the noise prediction model are thus optimized by minimizing the mean squared error loss:

$$\mathcal{L}_{\text{DDPM}} = \mathbb{E}_{\mathbf{x}_0,t,\boldsymbol{\epsilon}}\left[||\boldsymbol{\epsilon} - \boldsymbol{\epsilon}_\theta(\mathbf{x}_t, t, c)||^2\right] \tag{3}$$

**Noise Schedule**    When creating a noised example $\mathbf{x}_t$ (Equation 2), $\bar{\alpha}_t$ determines the variance of the noise for timestep $t$. In this work, we adopt the commonly used cosine scheduler Nichol & Dhariwal (2021), which largely follows $\sqrt{\bar{\alpha}_t} \approx \cos(\frac{t}{T} \cdot \frac{\pi}{2})$ with some adjustments.

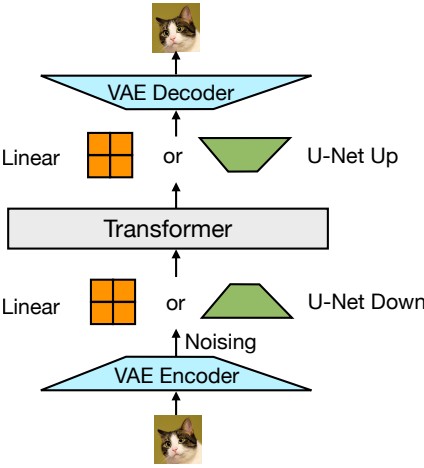

Figure 3: We convert images to and from latent representations using a pretrained VAE, and then into patch representations with either a simple linear layer or U-Net down blocks.

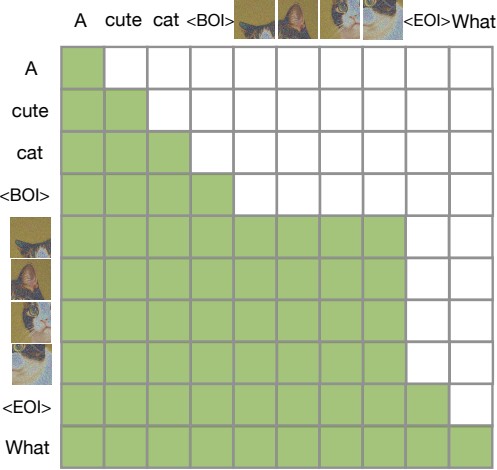

Figure 4: Expanding on the causal mask, Transfusion allows patches of the same image to condition on each other.

**Inference**   Decoding is done iteratively, pealing away some of the noise at each step. Starting with pure Gaussian noise at $\mathbf{x}_T$, the model $\boldsymbol{\epsilon}_\theta(\mathbf{x}_t, t, c)$ predicts the noise accumulated at timestep $t$. The predicted noise is then scaled according to the noise schedule, and the proportional amount of predicted noise is removed from $\mathbf{x}_t$ to produce $\mathbf{x}_{t-1}$. In practice, inference is done over fewer timesteps than training. Classifier-free guidance (CFG) (Ho & Salimans, 2022) is often used to improve generation by contrasting the prediction of the model conditioned on the context $c$ with the unconditioned prediction, at the cost of doubling the computation.

## 2.3   LATENT IMAGE REPRESENTATION

Early diffusion models worked directly in pixel space (Ho et al., 2020), but this proved computationally expensive. Variational autoencoders (VAEs) (Kingma & Welling, 2013) can save compute by encoding images into a lower-dimensional latent space. Implemented as deep CNNs, modern VAEs are trained on a combination of reconstruction and regularization losses (Esser et al., 2021), allowing downstream models like latent diffusion models (LDMs) (Rombach et al., 2022a) to operate efficiently on compact image patch embeddings; e.g. represent every $8\times8$ pixel patch as an 8-dimensional vector. For autoregressive language modeling approaches (Ramesh et al., 2021; Yu et al., 2022), images must be discretized. Discrete autoencoders, such as vector-quantized VAEs (VQ-VAE) (Van Den Oord et al., 2017), achieve this by introducing a quantization layer (and related regularization losses) that maps continuous latent embeddings to discrete tokens.

## 3   TRANSFUSION

Transfusion is a method for training a single unified model to understand and generate both discrete and continuous modalities. Our main innovation is demonstrating that we can use separate losses for different modalities – language modeling for text, diffusion for images – over shared data and parameters. Figure 1 illustrates Transfusion.

**Data Representation**   We experiment with data spanning two modalities: discrete text and continuous images. Each text string is tokenized into a sequence of discrete tokens from a fixed vocabulary, where each token is represented as an integer. Each image is encoded as latent patches using a VAE (see §2.3), where each patch is represented as a continuous vector; the patches are sequenced left-to-right top-to-bottom to create a sequence of patch vectors from each image. For mixed-modal examples, we surround each image sequence with special *beginning of image* (BOI) and *end of image* (EOI) tokens before inserting it to the text sequence; thus, we arrive at a single sequence potentially containing both discrete elements (integers representing text tokens) and continuous elements (vectors representing image patches).

**Model Architecture** The vast majority of the model's parameters belong to a single transformer, which processes every sequence, regardless of modality. The transformer takes a sequence of high-dimensional vectors in $\mathbb{R}^d$ as input, and produces similar vectors as output. To convert our data into this space, we use lightweight modality-specific components with unshared parameters. For text, these are the embedding matrices, converting each input integer to vector space and each output vector into a discrete distribution over the vocabulary. For images, we experiment with two alternatives for compressing local windows of $k \times k$ patch vectors into a single transformer vector (and vice versa): (1) a simple linear layer, and (2) up and down blocks of a U-Net (Nichol & Dhariwal, 2021; Saharia et al., 2022). Figure 3 illustrates the overall architecture.

**Transfusion Attention** Language models typically use causal masking to efficiently compute the loss and gradients over an entire sequence in a single forward-backward pass without leaking information from future tokens. While text is naturally sequential, images are not, and are usually modeled with unrestricted (bidirectional) attention. Transfusion combines both attention patterns by applying causal attention to every element in the sequence, and bidirectional attention within the elements of each individual image. This allows every image patch to attend to every other patch within the same image, but only attend to text or patches of other images that appeared previously in the sequence. We find that enabling intra-image attention significantly boosts model performance (see §4.3). Figure 4 shows an example Transfusion attention mask.

**Training Objective** To train our model, we apply the language modeling objective $\mathcal{L}_{\text{LM}}$ to predictions of text tokens and the diffusion objective $\mathcal{L}_{\text{DDPM}}$ to predictions of image patches. LM loss is computed per token, while diffusion loss is computed per image, which may span multiple elements (image patches) in the sequence. Specifically, we add noise $\epsilon$ to each input latent image $\mathbf{x}_0$ according to the diffusion process to produce $\mathbf{x}_t$ before patchification, and then compute the image-level diffusion loss.[1] We combine the two losses by simply adding the losses computed over each modality with a balancing coefficient $\lambda$:

$$\mathcal{L}_{\text{Transfusion}} = \mathcal{L}_{\text{LM}} + \lambda \cdot \mathcal{L}_{\text{DDPM}} \tag{4}$$

This formulation is a specific instantiation of a broader idea: combining a discrete distribution loss with a continuous distribution loss to optimize the same model. We leave further exploration of this space, such as replacing diffusion with flow matching (Lipman et al., 2022)), to future work.

**Inference** Reflecting the training objective, our decoding algorithm also switches between two modes: LM and diffusion. In *LM mode*, we follow the standard practice of sampling token by token from the predicted distribution. When we sample a BOI token, the decoding algorithm switches to *diffusion mode*, where we follow the standard procedure of decoding from diffusion models. Specifically, we append a pure noise $\mathbf{x}_T$ in the form of $n$ image patches to the input sequence (depending on the desired image size), and denoise over $T$ steps. At each step $t$, we take the noise prediction and use it to produce $\mathbf{x}_{t-1}$, which then overwrites $\mathbf{x}_t$ in the sequence; i.e. the model always conditions on the last timestep of the noised image and cannot attend to previous timesteps. Once the diffusion process has ended, we append an EOI token to the predicted image, and switch back to LM mode. This algorithm enables the generation of any mixture of text and image modalities.

## 4 EXPERIMENTS

We demonstrate in a series of controlled experiments that Transfusion is a viable, scalable method for training a unified multi-modal model. The setup of our experiments is detailed in Appendix B.1.

### 4.1 SETUP

**Evaluation** We evaluate model performance on a collection of standard uni-modal and cross-modal benchmarks (Table 7 in Appendix). For text-to-text, we measure perplexity on 20M held-out tokens from Wikipedia and the C4 corpus (Raffel et al., 2019), as well as accuracy on the pretraining evaluation suite of Llama 2 (Touvron et al., 2023b). For text-to-image, we use the MS-COCO benchmark (Lin et al., 2014), where we generate images on randomly selected 30k prompts from validation set and measure their photo-realism using zero-shot Frechet Inception Distance (FID)

---

[1]Ergo, downstream tokens condition on noisy images during training. See §B.2 for further discussion.

(Heusel et al., 2017) as well as their alignment with the prompts using CLIP score (Radford et al., 2021).[2] We also evaluate the model's ability to generate image captions; we report CIDEr (Vedantam et al., 2015) scores on the Karpathy test split of MS-COCO (Lin et al., 2014). These evaluations provide signal for investigation of scaling laws (§4.2) and ablations (§4.3). To compare with recent work in diffusion models, we evaluate our largest scale model (§4.4) also on GenEval (Ghosh et al., 2023), a benchmark that examines a model's ability to generate an accurate depiction of the prompt.

**Baseline** At the time of writing, the prominent open-science method for training a single mixed-modal model that can generate both text and images is to quantize images into discrete tokens, and then model the entire token sequence with a standard language model (Ramesh et al., 2021; Yu et al., 2022; 2023). We follow the recipe of Chameleon (Chameleon Team, 2024) to train a family of data- and compute-controlled baseline models, which we can directly compare to our Transfusion models. The key difference between Chameleon and Transfusion is that while Chameleon discretizes images and processes them as tokens, Transfusion keeps images in continuous space, removing the quantization information bottleneck. To further minimize any confounding variables, we train the VAEs for Chameleon and Transfusion using exactly the same data, compute, and architecture, with the only differentiator being the quantization layer and codebook loss of Chameleon's VQ-VAE (see details below). Chameleon also deviates from the Llama transformer architecture, adding query-key normalization, post-normalization, denominator loss, and a lower learning rate of 1e-4 to manage training instability, which incur an efficiency cost (see §4.2).[3]

**Data** For almost all of our experiments, we sample 0.5T tokens (patches) from two datasets at a 1:1 token ratio. For text, we use the Llama 2 tokenizer and corpus (Touvron et al., 2023b) of 2T tokens. For images, we use a collection of 380M licensed Shutterstock images and captions. Each image is center-cropped and resized to produce a 256×256 pixel image. We randomly order the image and captions, ordering the caption first 80% of the time.

In one experiment (4.4) we scale up the total training data to 2T tokens (1T text tokens and about 3.5B caption-image pairs at 256 patches per image). To diversify, we add 220M publicly available images with captions, prefiltered to not contain people. To rebalance the distribution, we upsample 80M Shutterstock images containing people. We also add data from Conceptual 12M (CC12M) (Changpinyo et al., 2021), reaching a total mixture of 692M image-caption pairs per epoch. Finally, we upweight the portion of high-aesthetic images in the last 1% of the training schedule.

**Latent Image Representation** We train a 86M parameter VAE following Esser et al. (2021). We use a CNN encoder and decoder, and latent dimension 8. The training objective is combines reconstruction and regularization losses ( Appendix C) For VQ-VAE training, we follow the same setup described for VAE training, except we replace $\mathcal{L}_{KL}$ with the standard codebook commitment loss with $\beta = 0.25$ (Van Den Oord et al., 2017). We use a codebook of 16,384 token types.

**Model Configuration** To investigate scaling trends, we train models at five different sizes – 0.16B, 0.37B, 0.76B, 1.4B, and 7B parameters – following the standard settings from Llama (Touvron et al., 2023a) (Table 8 in Appendix). In configurations that use linear patch encoding (§4.2 and §4.3), the number of additional parameters is insignificant, accounting for fewer than 0.5% of total parameters in every configuration. When using U-Net patch encoding (§4.3 and §4.4), these parameters add up to 0.27B additional parameters across all configurations; while this is a substantial addition of parameters to smaller models, these layers amount to only a 3.8% increase of the 7B configuration, almost identical to the number of parameters in the embedding layers.

**Optimization** We use AdamW ($\beta_1 = 0.9$, $\beta_2 = 0.95$, $\epsilon = $1e-8) with a learning rate of 3e-4, warmed up for 4000 steps and decaying to 1.5e-5 using a cosine scheduler. We train on sequences of 4096 tokens in batches of 2M tokens for 250k steps, reaching 0.5T tokens in total. In our large-scale experiment (§4.4), we train with a batch size of 4M tokens over 500k steps, totalling 2T tokens. We set the $\lambda$ coefficient in the Transfusion objective (Equation 4) to 5 following preliminary experiments.

**Inference** In text mode, we use greedy decoding for generating text. Ranked classification is used for the Llama evaluation suite. For image generation, we follow the standard of 250 diffusion steps (the model is trained on 1,000 timesteps). We follow Chameleon and use CFG with a coefficient of 5

---

[2]We follow common practice for ablations and use only 5k examples to compute FID and CLIP in §4.3.

[3]Removing these deviations in preliminary experiments encountered optimization instabilities in Chameleon.

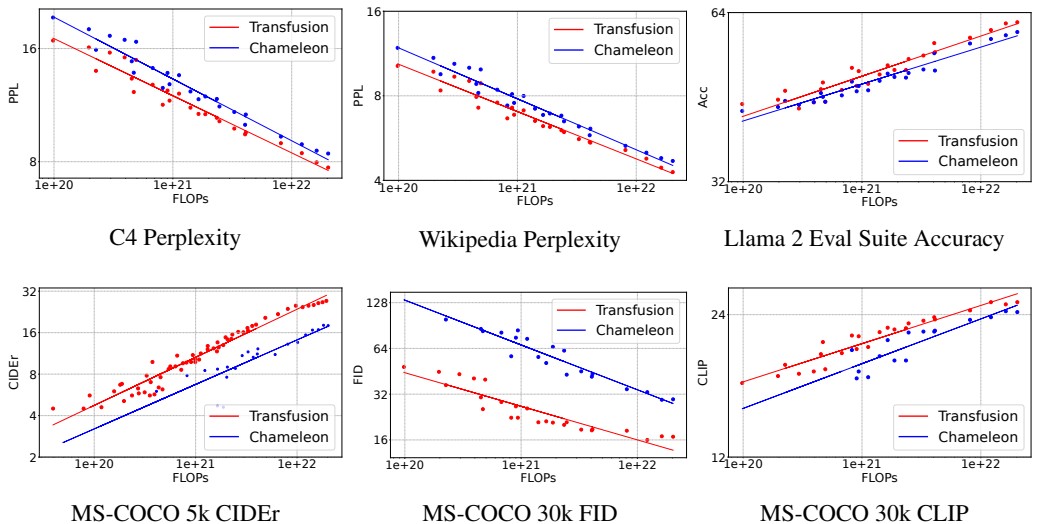

Figure 5: Performance of Transfusion and Chameleon models at different scales, controlled for parameters, data, and compute. All axes are logarithmic.

in the controlled comparison experiments (§4.2). This value is suboptimal for Transfusion, and so we use a CFG coefficient of 3 throughout the ablation experiments (§4.3), and follow the standard practice of tuning the coefficient for each benchmark in our large scale experiment (§4.4).

## 4.2 CONTROLLED COMPARISON WITH CHAMELEON

We run a series of controlled experiments to compare Transfusion with Chameleon at different model sizes ($N$) and token counts ($D$), using the combination of both as a proxy for FLOPs ($6ND$). For simplicity and parameter control, the Transfusion variant in these experiments uses simple linear image encoder/decoder with patch size 2×2, as well as bidirectional attention. For each benchmark, we plot all results on a log-metric over log-FLOPs curve and regress linear trendlines. We also estimate relative compute efficiency by measuring the parity FLOP ratio: the ratio between the number of FLOPs required by Transfusion and Chameleon to reach the same level of performance.

Figure 5 visualizes the scaling trends, and Table 1 shows the results of the largest models in this controlled setting and their estimated parity FLOP ratio. In every benchmark, Transfusion consistently exhibits better scaling laws than Chameleon. While the lines are close to parallel, there is a significant gap in Transfusion's favor. The difference in compute efficiency is particularly striking in image generation, where FID Transfusion achieves parity with Chameleon using 34× less compute.

Surprisingly, text-only benchmarks also reveal better performance with Transfusion, even though both Transfusion and Chameleon model text in the same way. We investigate this phenomenon by ablating the various changes leading up to Transfusion and Chameleon from the original Llama 2 recipe. Table 2 shows that while Transfusion does come at a non-zero cost to text performance, the Chameleon recipe suffers from both the stability modifications made to the architecture and from the introduction of image tokens. Training on quantized image tokens degrades text performance

| Model | C4
PPL ($\downarrow$) | Wiki
PPL ($\downarrow$) | Llama
Acc ($\uparrow$) | MS-COCO
CDr ($\uparrow$) | FID ($\downarrow$) | CLIP ($\uparrow$) |
|---|---|---|---|---|---|---|
| Transfusion | **7.72** | **4.28** | **61.5** | **27.2** | **16.8** | **25.5** |
| Chameleon | 8.41 | 4.69 | 59.1 | 18.0 | 29.6 | 24.3 |
| Parity FLOP Ratio | 0.489 | 0.526 | 0.600 | 0.218 | 0.029 | 0.319 |

Table 1: Performance of the largest (7B) Transfusion and Chameleon models in a controlled setting. Both models were trained on 0.5T tokens. **Parity FLOP Ratio** is the relative amount of Transfusion FLOPs needed to match the results of Chameleon 7B.

| Model | | Batch | C4 PPL ($\downarrow$) | Wiki PPL ($\downarrow$) | Llama Acc ($\uparrow$) |
|---|---|---|---|---|---|
| Llama 2 | | 1M Text Tokens | 10.1 | 5.8 | 53.7 |
| Transfusion | + Diffusion | + 1M Image Patches | (+0.3) 10.4 | (+0.2) 6.0 | (-1.0) 52.7 |
| Chameleon | + Stability Modifications | 1M Text Tokens | (+0.9) 11.0 | (+0.5) 6.3 | (-1.8) 51.9 |
| | + LM Loss on Image Tokens | + 1M Image Tokens | (+0.8) 11.8 | (+0.5) 6.8 | (-3.0) 48.9 |

Table 2: Performance of the 0.76B Transfusion and Chameleon models on text-only benchmarks, compared to the original Llama 2 recipe.

| Enc/Dec | Attention | C4 PPL ($\downarrow$) | Wiki PPL ($\downarrow$) | Llama Acc ($\downarrow$) | MS-COCO | | |
|---|---|---|---|---|---|---|---|
| | | | | | CDr ($\uparrow$) | FID ($\downarrow$) | CLIP ($\uparrow$) |
| Linear | Causal | 10.4 | 6.0 | 51.4 | 12.7 | 61.3 | 23.0 |
| | Bidirectional | 10.4 | 6.0 | 51.7 | 16.0 | 20.3 | 24.0 |
| U-Net | Causal | 10.3 | 5.9 | 52.0 | 23.3 | 16.8 | 25.3 |
| | Bidirectional | 10.3 | 5.9 | 51.9 | 25.4 | 16.7 | 25.4 |

Table 3: Performance of 0.76B Transfusion models with and without intra-image bidirectional attention. Patch size is set at 2×2 latent pixels.

more than diffusion on all three benchmarks. One hypothesis is that this stems from the competition between text and image tokens in the output distribution; alternatively, it is possible that diffusion is more efficient at image generation and requires fewer parameters, allowing Transfusion models to use more capacity than Chameleon to model text. We leave further investigation of this phenomenon to future research.

## 4.3 ARCHITECTURE ABLATIONS

We explore improvements and extensions that are applicable to Transfusion alone in this section.

### 4.3.1 ATTENTION MASKING

We first examine the necessity of intra-image bidirectional attention. Table 3 shows that enabling this attention pattern beyond the standard causal attention is advantageous throughout all benchmarks, and using both image encoding/decoding architectures. In particular, we notice a significant improvement in FID when using linear encoding layers (61.3→20.3). In the causal-only version of this architecture, there is no flow of information from patches that appear later in the sequence to those before; since U-Net blocks contain bidirectional attention within, independent of the transformer's attention mask, this gap is less pronounced when they are applied.

### 4.3.2 PATCH SIZE

Transfusion models can be defined over different sizes of latent pixel patches. Larger patch sizes allow the model to pack more images in each training batch and dramatically reduce inference compute, but may come at a performance cost. Table 4 and Table 9 in Appendix sheds light on these performance trade-offs. While performance does decrease consistently as each image is represented by fewer patches with linear encoding, models with U-Net encoding benefit from larger patches on tasks involving the image modality. We posit that this is due to the greater amount of total images (and diffusion noise) seen during training. We also observe that text performance deteriorates with larger patches, perhaps because transfusion needs to exert more resources (i.e. parameters) to learn how to process images with fewer patches and thus less inference compute.

### 4.3.3 PATCH ENCODING/DECODING ARCHITECTURE

Our experiments so far indicate an advantage to using the U-Net up and down blocks instead of a simple linear layer. One possible reason is that the model benefits from the inductive biases of the U-Net architecure; an alternative hypothesis is that this advantage stems from the significant increase in overall model parameters introduced by the U-Net layers. To decouple these two confounders, we scale up the core transformer to 7B parameters, while keeping the amount of U-Net parameters

| Enc/Dec | Latent/ Patch | Pixel/ Patch | Patch/ Image | C4 PPL (↓) | Wiki PPL (↓) | Llama Acc (↓) | MS-COCO | | |
|---|---|---|---|---|---|---|---|---|---|
| | | | | | | | CDr (↑) | FID (↓) | CLIP (↑) |
| None | 1×1 | 8×8 | 1024 | **10.3** | **5.9** | **52.2** | 12.0 | 21.0 | 24.0 |
| U-Net | 2×2 | 16×16 | 256 | **10.3** | **5.9** | 51.9 | 25.4 | 16.7 | 25.4 |
| | 4×4 | 32×32 | 64 | 10.7 | 6.2 | 50.7 | **29.9** | **16.0** | **25.7** |
| | 8×8 | 64×64 | 16 | 11.4 | 6.6 | 49.2 | 29.5 | 16.1 | 25.2 |

Table 4: Performance of 0.76B Transfusion models with different patch sizes. Bolded figures indicate global best, underlines indicate best within architecture.

| Model Params | Enc/Dec | Δ Enc/Dec Params | C4 PPL (↓) | Wiki PPL (↓) | Llama Acc (↑) | MS-COCO | | |
|---|---|---|---|---|---|---|---|---|
| | | | | | | CDr (↑) | FID (↓) | CLIP (↑) |
| 0.37B | Linear | 0.4% | 12.0 | 7.0 | 47.9 | 11.1 | 21.5 | 22.4 |
| | U-Net | 71.3% | 11.8 | 6.9 | 48.8 | 21.1 | 18.1 | 24.9 |
| 1.4B | Linear | 0.4% | 9.5 | 5.4 | 53.8 | 19.1 | 19.4 | 24.3 |
| | U-Net | 19.3% | 9.4 | 5.4 | 53.4 | 28.1 | 16.6 | 25.7 |
| 7B | Linear | 0.3% | 7.7 | 4.3 | 61.5 | 27.2 | 18.6 | 25.9 |
| | U-Net | 3.8% | 7.8 | 4.3 | 61.1 | 33.7 | 16.0 | 26.5 |

Table 5: Performance of linear and U-Net variants of Transfusion across different model sizes. Patch size is set at 2×2 latent pixels. Model parameters refers to the transformer alone.

(almost) constant; in this setting, the additional encoder/decoder parameters account for only a 3.8% increase of total model parameters, equivalent to the amount of token embedding parameters.

Table 5 Table 10 (Appendix) shows that even though the relative benefit of U-Net layers shrinks as the transformer grows, it does not diminish. In image generation, for example, the U-Net encoder/decoder allows much smaller models to obtain better FID scores than the 7B model with linear patchification layers. We observe a similar trend in image captioning, where adding U-Net layers boosts the CIDEr score of a 1.4B transformer (1.67B combined) beyond the performance of the linear 7B model. Overall, it appears that there are indeed inductive bias benefits to U-Net encoding and decoding of images beyond the mere addition of parameters.

## 4.4 COMPARISON WITH IMAGE GENERATION LITERATURE

Our experiments thus far have covered controlled comparisons with Chameleon and Llama, but we have yet to compare Transfusion's image generation capabilities to those of state-of-the-art image generation models. To that end, we train a 7B parameter model with U-Net encoding/decoding layers (2×2 latent pixel patches) over the equivalent of 2T tokens, comprising of 1T text corpus tokens and 3.5B images and their captions. While the Transfusion variant in §4.2 favored simplicity and experimental control, the design choices and data mixture (§4.1) of this variant lean a bit more towards image generation. Figure 7 and Appendix D showcase generated images from this model.

We compare the performance of our model to reported results of other similar scale image generation models, as well as some publicly available text generating models for reference. Table 6 shows that Transfusion achieves similar performance to high-performing image generation models such as DeepFloyd (Stability AI, 2024), while surpassing previously published models including SDXL (Podell et al., 2023). While Transfusion does lag behind SD 3 (Esser et al., 2024a), this model leveraged synthetic image captions through backtranslation (Betker et al., 2023), which enhances its GenEval performance by 6.5% absolute (0.433→0.498) at smaller scale; for simplicity, our experimental setup only included natural data. Finally, we note that our Transfusion model can also generate text, and performs on par with the Llama models, which were trained on the same text data distribution (§4.1).

## 4.5 IMAGE EDITING

Our Transfusion models, which have been pretrained on text-text, image-text, and text-image data, perform well across these modality pairings. Can these models extend their capabilities to generate

| Model | Model Params | Text Tokens | Images | Llama Acc (↑) | COCO FID (↓) | Gen Eval (↑) |
|---|---|---|---|---|---|---|
| Llama 1 (Touvron et al., 2023a) | 7.1B | 1.4T | — | 66.1 | — | — |
| Llama 2 (Touvron et al., 2023b) | 7.1B | 2.0T | — | 66.3 | — | — |
| Chameleon (Chameleon Team, 2024) | 7.1B | 6.0T | 5.0B | 67.1 | 26.74 | 0.39 |
| Imagen (Saharia et al., 2022) | 2.6B + 4.7B* | — | 5.0B | — | 7.27 | — |
| Parti (Yu et al., 2022) | 20B | — | 4.8B | — | ʳ7.23 | — |
| SD 2.1 (Rombach et al., 2022b) | 0.9B + 0.1B* | — | 2.3B | — | — | 0.50 |
| DALL-E 2 (Ramesh et al., 2022) | 4.2B + 1B* | — | 2.6B | — | 10.39 | 0.52 |
| SDXL (Podell et al., 2023) | 2.6B + 0.8B* | — | 1.6B | — | — | 0.55 |
| DeepFloyd (Stability AI, 2024) | 5.5B + 4.7B* | — | 7.5B | — | 6.66 | 0.61 |
| SD 3 (Esser et al., 2024b) | 8B + 4.7B* | — | ˢ2.0B | — | — | 0.68 |
| Transfusion (Ours) | 7.3B | 1.0T | 3.5B | 66.1 | 6.78 | 0.63 |

Table 6: Performance of a 7B Transfusion model (U-Net encoder/decoder layers, 2×2 latent pixel patches) trained on the equivalent of 2T tokens, compared to similar scale models in the literature. Except Chameleon, all the other models are restricted to generating one modality (either text or image). * Frozen text encoder parameters. ʳ Parti samples 16 images for every prompt and then reranks with an auxiliary scoring model. ˢ SD 3 trains with synthetic caption data, which provides boosts GenEval performance.

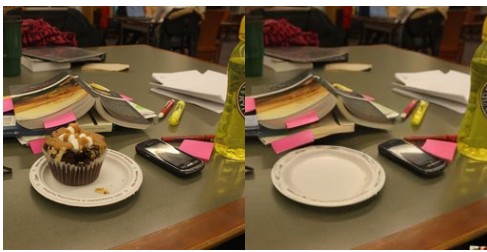

Remove the cupcake on the plate.

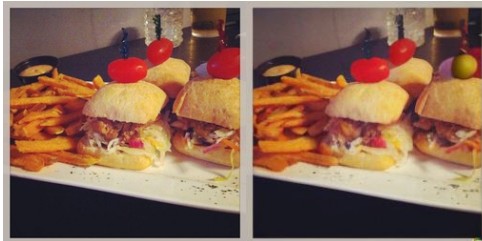

Change the tomato on the right to a green olive.

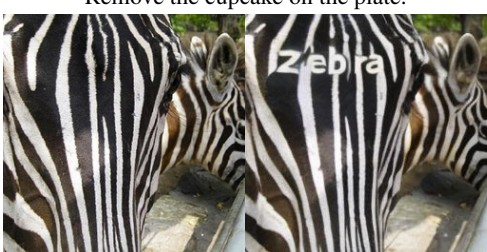

Write the word "Zebra" in Arial bold.

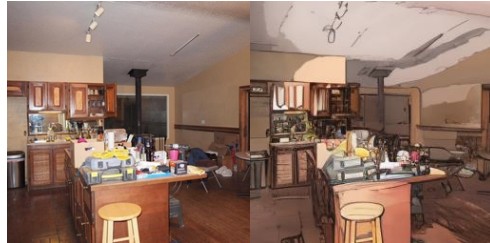

Change this to cartoon style.

Figure 6: Edited images from a fine-tuned 7B Transfusion model.

images based on other images? To investigate, we fine-tuned our 7B model (§4.4) using a dataset of only 8k publicly available image editing examples, where each example consists of an input image, an edit prompt, and an output image. This approach, inspired by LIMA (Zhou et al., 2024), allows us to assess how well the model can generalize to image-to-image generation, a scenario not covered during pretraining. Manual examination of random examples from the EmuEdit test set (Sheynin et al., 2024), shown in Figure 6 and Appendix 4.5, reveals that our fine-tuned Transfusion model performs image edits as instructed. Despite the limitations of this experiment, the findings suggest that Transfusion models can indeed adapt to and generalize across new modality combinations. We leave further exploration of this promising direction to future research.

## 5 CONCLUSION

This work explores how to bridge the gap between the state of the art in discrete sequence modeling (next token prediction) and continuous media generation (diffusion). We propose a simple, yet previously unexplored solution: train a single joint model on two objectives, tying each modality to its preferred objective. Our experiments show that Transfusion scales efficiently, incurring little to no parameter sharing cost, while enabling the generation of any modality.

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

| Input | Output | Benchmark | Metric |
|-------|--------|-----------|--------|
| Text | Text | Wikipedia
C4
Llama 2 Eval Suite | Perplexity ($\downarrow$)
Perplexity ($\downarrow$)
Accuracy ($\uparrow$) |
| Image | Text | MS-COCO 5k | CIDEr ($\uparrow$) |
| Text | Image | MS-COCO 30k
GenEval | FID ($\downarrow$), CLIP ($\uparrow$)
GenEval score ($\uparrow$) |

| Size | Layers | Emb Dim | Att Heads |
|------|--------|---------|-----------|
| 0.16B | 16 | 768 | 12 |
| 0.37B | 24 | 1024 | 16 |
| 0.76B | 24 | 1536 | 24 |
| 1.4B | 24 | 2048 | 16 |
| 7B | 32 | 4096 | 32 |

Table 7: An overview of the evaluation suite used in this work.

Table 8: Model sizes and configurations for both Transfusion and baselines.

## A    RELATED WORK

Most existing multi-modal models are built on the idea of attaching two or more modality-specific architectures together, often pretraining each component separately in advance. State-of-the-art image and video generation models, for instance, use large pretrained text encoders to represent their input prompts in latent space, which can then be used to condition diffusion models (Saharia et al., 2022). In fact, recent work fuses representations from multiple off-the-shelf encoders to enhance performance (Podell et al., 2023; Esser et al., 2024b). A similar pattern can be observed in the vision language model literature, where typically a pretrained language model is complemented by pretrained modality-specific encoders/decoders via projection layers to/from the pretrained text space. Examples include Flamingo (Alayrac et al., 2022) and LLaVA (Liu et al., 2024) for visual understanding, GILL (Koh et al., 2024) for visual generation, and DreamLLM (Dong et al., 2024) for both visual comprehension and generation. In contrast, Transfusion has one unified architecture learned end-to-end to generate both text and images.

Prior work on end-to-end multi-modal models includes examples such as Fuyu (Bavishi et al., 2023), which uses image patches as inputs for visual understanding, and Chameleon (Chameleon Team, 2024), which converts each image to a sequence of discretized tokens and then trains over the combined text-image token sequences. However, these approaches are either restricted to input-level multi-modal tasks, or lag behind state-of-the-art models (i.e. diffusion models) in continuous data generation. Transfusion provides a simple, end-to-end solution to multi-modal learning that understands and generates high-quality multi-modal data.

An interesting area of recent acrive research is the application diffusion models and their generalizations to discrete text generation (Li et al., 2022; Gat et al., 2024). However, this approach has yet to achieve the performance and scale of standard autoregressive language models. Future research in this direction may unlock new ways to fuse discrete and continuous modalities in a single model.

## B    EXPERIMENTS

### B.1    SETUP

**Evaluation** We evaluate model performance on a collection of standard uni-modal and cross-modal benchmarks (Table 7). For text-to-text, we measure perplexity on 20M held-out tokens from Wikipedia and the C4 corpus (Raffel et al., 2019), as well as accuracy on the pretraining evaluation suite of Llama 2 (Touvron et al., 2023b).[4] For text-to-image, we use the MS-COCO benchmark (Lin et al., 2014), where we generate images on randomly selected 30k prompts from validation set and measure their photo-realism using zero-shot Frechet Inception Distance (FID) (Heusel et al., 2017) as well as their alignment with the prompts using CLIP score (Radford et al., 2021).[5] We also evaluate the model's ability to generate image captions; we report CIDEr (Vedantam et al., 2015) scores on the Karpathy test split of MS-COCO (Lin et al., 2014). These evaluations provide signal for

---

[4]The Llama 2 evaluation suite includes HellaSwag (Zellers et al., 2019), PIQA (Bisk et al., 2020), SIQA (Sap et al., 2019), WinoGrande (Sakaguchi et al., 2021), ARC-e and -c (Clark et al., 2018), and BoolQ (Clark et al., 2019). We report the average 0-shot task accuracy on these benchmarks.

[5]We follow common practice for ablations and use only 5k examples to compute FID and CLIP in §4.3.

investigation scaling laws (§4.2) and ablations (§4.3). To compare with recent literature in diffusion models, we evaluate our largest scale model (§4.4) also on GenEval (Ghosh et al., 2023), a benchmark that examines a model's ability to generate an accurate depiction of the prompt.

**Baseline**    At the time of writing, the prominent open-science method for training a single mixed-modal model that can generate both text and images is to quantize images into discrete tokens, and then model the entire token sequence with a standard language model (Ramesh et al., 2021; Yu et al., 2022; 2023). We follow the recipe of Chameleon (Chameleon Team, 2024) to train a family of data- and compute-controlled baseline models, which we can directly compare to our Transfusion models. The key difference between Chameleon and Transfusion is that while Chameleon discretizes images and processes them as tokens, Transfusion keeps images in continuous space, removing the quantization information bottleneck. To further minimize any confounding variables, we train the VAEs for Chameleon and Transfusion using exactly the same data, compute, and architecture, with the only differentiator being the quantization layer and codebook loss of Chameleon's VQ-VAE (see details below). Chameleon also deviates from the Llama transformer architecture, adding query-key normalization, post-normalization, denominator loss, and a lower learning rate of 1e-4 to manage training instability, which incur an efficiency cost (see §4.2).[6]

**Data**    For almost all of our experiments, we sample 0.5T tokens (patches) from two datasets at a 1:1 token ratio. For text, we use the Llama 2 tokenizer and corpus (Touvron et al., 2023b), containing 2T tokens across a diverse distribution of domains. For images, we use a collection of 380M licensed Shutterstock images and captions. Each image is center-cropped and resized to produce a $256{\times}256$ pixel image.[7] We randomly order the image and captions, ordering the caption first 80% of the time.

In one experiment (4.4) we scale up the total training data to 2T tokens (1T text tokens and about 3.5B caption-image pairs at 256 patches per image). To diversify, we add 220M publicly available images with captions, prefiltered to not contain people. To rebalance the distribution, we upsample 80M Shutterstock images containing people. We also add data from Conceptual 12M (CC12M) (Changpinyo et al., 2021), reaching a total mixture of 692M image-caption pairs per epoch. Finally, we upweight the portion of high-aesthetic images in the last 1% of the training schedule.

**Latent Image Representation**    We train a 86M parameter VAE following Esser et al. (2021). We use a CNN encoder and decoder, and latent dimension 8. The training objective is combines reconstruction and regularization losses.[8] Our implementation reduces an image of $256{\times}256$ pixels to a $32{\times}32{\times}8$ tensor, where each latent 8-dimensional latent pixel represents (conceptually) an $8{\times}8$ pixel patch in the original image, and trains for 1M steps. For VQ-VAE training, we follow the same setup described for VAE training, except we replace $\mathcal{L}_{KL}$ with the standard codebook commitment loss with $\beta = 0.25$ (Van Den Oord et al., 2017). We use a codebook of 16,384 token types.

**Model Configuration**    To investigate scaling trends, we train models at five different sizes – 0.16B, 0.37B, 0.76B, 1.4B, and 7B parameters – following the standard settings from Llama (Touvron et al., 2023a). Table 8 describes each setting in detail. In configurations that use linear patch encoding (§4.2 and §4.3), the number of additional parameters is insignificant, accounting for fewer than 0.5% of total parameters in every configuration. When using U-Net patch encoding (§4.3 and §4.4), these parameters add up to 0.27B additional parameters across all configurations; while this is a substantial addition of parameters to smaller models, these layers amount to only a 3.8% increase of the 7B configuration, almost identical to the number of parameters in the embedding layers.

**Optimization**    We randomly initialize all model parameters, and optimize them using AdamW ($\beta_1 = 0.9$, $\beta_2 = 0.95$, $\epsilon = $1e-8) with a learning rate of 3e-4, warmed up for 4000 steps and decaying to 1.5e-5 using a cosine scheduler. We train on sequences of 4096 tokens in batches of 2M tokens for 250k steps, reaching 0.5T tokens in total. In our large-scale experiment (§4.4), we train with a batch size of 4M tokens over 500k steps, totalling 2T tokens. We regularize with weight decay of 0.1 and

---

[6]Removing these deviations in preliminary experiments encountered optimization instabilities in Chameleon.

[7]Depending on the compression rate of the patch encoder (see Model Architecture in §3), each image will be represented by either 1024, 256, 64, or 16 elements in the sequence. Since the text/image ratio is constant during training, higher compression rates enable training on more images in total, at the cost of less compute per image.

[8]See Appendix C for details.

| Enc/Dec | Latent/ Patch | Pixel/ Patch | Patch/ Image | C4 PPL (↓) | Wiki PPL (↓) | Llama Acc (↓) | MS-COCO | | |
|---|---|---|---|---|---|---|---|---|---|
| | | | | | | | CDr (↑) | FID (↓) | CLIP (↑) |
| None | 1×1 | 8×8 | 1024 | **10.3** | **5.9** | **52.2** | 12.0 | 21.0 | 24.0 |
| Linear | 2×2 | 16×16 | 256 | 10.4 | 6.0 | 51.7 | 16.0 | 20.3 | 24.0 |
| | 4×4 | 32×32 | 64 | 10.9 | 6.3 | 49.8 | 14.3 | 25.6 | 22.6 |
| | 8×8 | 64×64 | 16 | 11.7 | 6.9 | 47.7 | 11.3 | 43.5 | 18.9 |
| U-Net | 2×2 | 16×16 | 256 | **10.3** | **5.9** | 51.9 | 25.4 | 16.7 | 25.4 |
| | 4×4 | 32×32 | 64 | 10.7 | 6.2 | 50.7 | **29.9** | **16.0** | **25.7** |
| | 8×8 | 64×64 | 16 | 11.4 | 6.6 | 49.2 | 29.5 | 16.1 | 25.2 |

Table 9: Performance of 0.76B Transfusion models with different patch sizes. Bolded figures indicate global best, underlines indicate best within architecture.

| Model Params | Enc/Dec | Δ Enc/Dec Params | C4 PPL (↓) | Wiki PPL (↓) | Llama Acc (↑) | MS-COCO | | |
|---|---|---|---|---|---|---|---|---|
| | | | | | | CDr (↑) | FID (↓) | CLIP (↑) |
| 0.16B | Linear | 0.5% | 14.8 | 8.8 | 44.2 | 6.2 | 37.6 | 20.0 |
| | U-Net | 106.1% | 14.4 | 8.5 | 45.7 | 15.3 | 18.8 | 23.9 |
| 0.37B | Linear | 0.4% | 12.0 | 7.0 | 47.9 | 11.1 | 21.5 | 22.4 |
| | U-Net | 71.3% | 11.8 | 6.9 | 48.8 | 21.1 | 18.1 | 24.9 |
| 0.76B | Linear | 0.4% | 10.4 | 6.0 | 51.7 | 16.0 | 20.3 | 24.0 |
| | U-Net | 35.5% | 10.3 | 5.9 | 51.9 | 25.4 | 16.7 | 25.4 |
| 1.4B | Linear | 0.4% | 9.5 | 5.4 | 53.8 | 19.1 | 19.4 | 24.3 |
| | U-Net | 19.3% | 9.4 | 5.4 | 53.4 | 28.1 | 16.6 | 25.7 |
| 7B | Linear | 0.3% | 7.7 | 4.3 | 61.5 | 27.2 | 18.6 | 25.9 |
| | U-Net | 3.8% | 7.8 | 4.3 | 61.1 | 33.7 | 16.0 | 26.5 |

Table 10: Performance of linear and U-Net variants of Transfusion across different model sizes. Patch size is set at 2×2 latent pixels. Model parameters refers to the transformer alone.

clip gradients by norm (1.0). We set the $\lambda$ coefficient in the Transfusion objective (Equation 4) to 5 following preliminary experiments; we leave further tuning of $\lambda$ to future work.

**Inference** In text mode, we use greedy decoding for generating text. Ranked classification is used for the Llama evaluation suite. For image generation, we follow the standard of 250 diffusion steps (the model is trained on 1,000 timesteps). We follow Chameleon and use CFG with a coefficient of 5 in the controlled comparison experiments (§4.2). This value is suboptimal for Transfusion, and so we use a CFG coefficient of 3 throughout the ablation experiments (§4.3), and follow the standard practice of tuning the coefficient for each benchmark in our large scale experiment (§4.4).

## B.2 IMAGE NOISING

Our experiments order 80% of image-caption pairs with the caption first, and the image conditioning on the caption, following the intuition that image generation may be a more data-hungry task than image understanding. The remaining 20% of the pairs condition the caption on the image. However, these images are noised as part of the diffusion objective. We thus measure the effect of limiting the diffusion noise to a maximum of $t = 500$ (half of the noise schedule) in the 20% of cases where images appear before their captions. Table 11 shows that noise limiting significantly improves image captioning, as measure by CIDEr, while having a relatively small effect (less than 1%) on other benchmarks.

| Model Params | Noise Limit | C4 PPL ($\downarrow$) | Wiki PPL ($\downarrow$) | Llama Acc ($\uparrow$) | MS-COCO | | |
| --- | --- | --- | --- | --- | --- | --- | --- |
| | | | | | CDr ($\uparrow$) | FID ($\downarrow$) | CLIP ($\uparrow$) |
| 0.76B | | 10.3 | 5.9 | 51.9 | 25.4 | 16.7 | 25.4 |
| | $\checkmark$ | 10.3 | 5.9 | 52.1 | **29.4** | 16.5 | 25.4 |
| 7B | | 7.8 | 4.3 | 61.1 | 33.7 | 16.0 | 26.5 |
| | $\checkmark$ | 7.7 | 4.3 | 60.9 | **35.2** | 15.7 | 26.3 |

Table 11: Performance of Transfusion with and without limiting the amount of sampled diffusion noise to a maximum of $t = 500$ when images appear before the caption. The models are U-Net variants encoding $2 \times 2$ latent pixel patches. Metrics that change by over 1% are bolded.

## C  AUTOENCODER DETAILS

The training objective for our VAE closely follows that of Esser et al. (2021):

$$\mathcal{L}_{\text{VAE}} = \mathcal{L}_1 + \mathcal{L}_{\text{LPIPS}} + 0.5\mathcal{L}_{\text{GAN}} + 0.2\mathcal{L}_{\text{ID}} + 0.000001\mathcal{L}_{\text{KL}}$$

where $L_1$ is L1 loss in pixel space, $L_{\text{LPIPS}}$ is perceptual loss based on LPIPS similarity Zhang et al. (2018), $L_{GAN}$ is a patch-based discriminator loss, $L_{\text{ID}}$ is a perceptual loss based on internal features of the Moco v2 model Chen et al. (2020), and $L_{\text{KL}}$ is the standard KL-regularization term to encourage encoder outputs towards a normal distribution. We delay the beginning of GAN training (i.e. including the adversarial loss in the loss function) to 50,000 steps, in order to let the VAE achieve sufficiently good reconstruction performance. We use a latent dimension of 8.

The training objective for the VQ-GAN matches that of the VAE, with one notable exception: we replace the $\mathcal{L}_{\text{KL}}$ loss with the standard codebook commitment loss $\mathcal{L}_{\text{codebook}}$ (Van Den Oord et al., 2017), which encourages encoder outputs and codebook vectors to be close together. We use $\beta = 0.25$, and use loss weighting 1.0. The final loss function for the VQ-VAE is therefore:

$$\mathcal{L}_{\text{VQ-VAE}} = \mathcal{L}_1 + \mathcal{L}_{\text{LPIPS}} + 0.5\mathcal{L}_{\text{GAN}} + 0.2\mathcal{L}_{\text{ID}} + \mathcal{L}_{\text{codebook}}$$

The vector quantization layer is applied after projecting the encoder outputs to 8-dimensional space. Outside of the loss function change and the quantization layer, the training setup for the VAE (for Transfusion) and VQ-VAE (for Chameleon) are the same (e.g. same amount of training compute, same training data, and same encoder/decoder architecture).

## D  EXAMPLES: IMAGE GENERATION

Figure 8 and Figure 9 show examples of images generated from a 7B Transfusion model trained on 2T multi-modal tokens (§4.4).

## E  EXAMPLES: IMAGE EDITING

Figure 10 show random examples of image editing by a fine-tuned 7B Transfusion model.

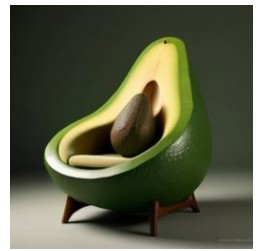

An armchair in the shape of an avocado

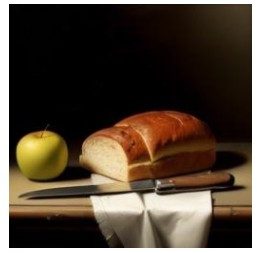

A bread, an apple, and a knife on a table

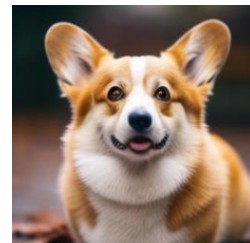

A corgi.

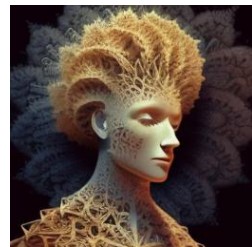

human life depicted entirely out of fractals

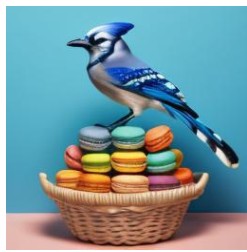

A blue jay standing on a large basket of rainbow macarons.

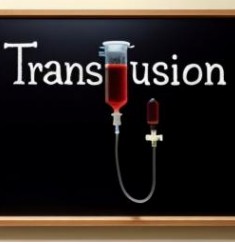

"Transfusion" is written on the blackboard.

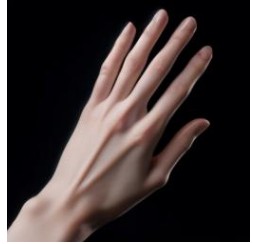

A close up photo of a human hand, hand model. High quality

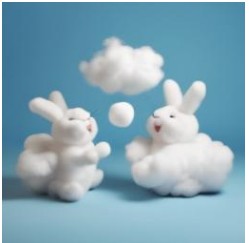

A cloud in the shape of two bunnies playing with a ball. The ball is made of clouds too.

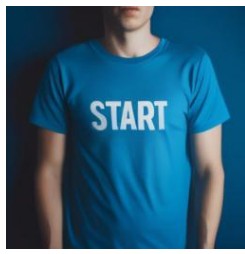

the word 'START' on a blue t-shirt

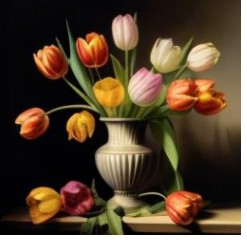

A Dutch still life of an arrangement of tulips in a fluted vase. The lighting is subtle, casting gentle highlights on the flowers and emphasizing their delicate details and natural beauty.

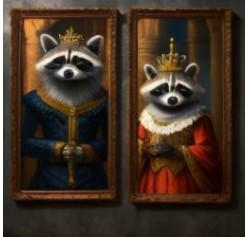

A wall in a royal castle. There are two paintings on the wall. The one on the left a detailed oil painting of the royal raccoon king. The one on the right a detailed oil painting of the royal raccoon queen.

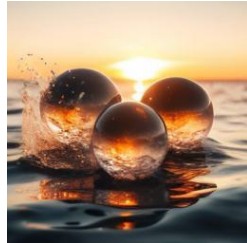

Three spheres made of glass falling into ocean. Water is splashing. Sun is setting.

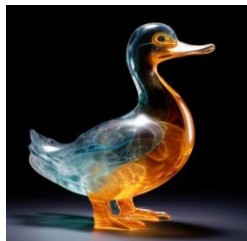

A transparent sculpture of a duck made out of glass.

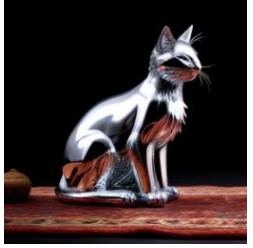

A chromeplated cat sculpture placed on a Persian rug.

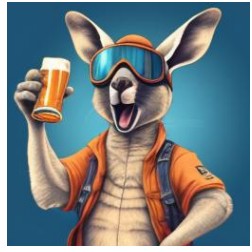

A kangaroo holding a beer, wearing ski goggles and passionately singing silly songs.

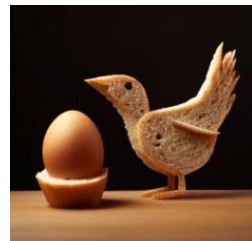

an egg and a bird made of wheat bread

Figure 7: Generated images from a 7B Transfusion trained on 2T multi-modal tokens.

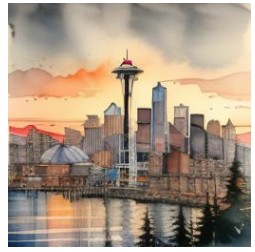
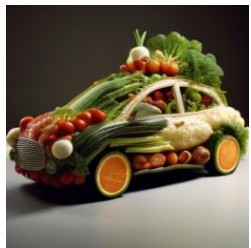
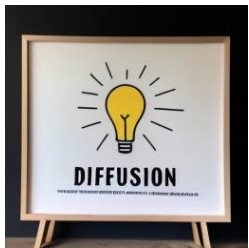
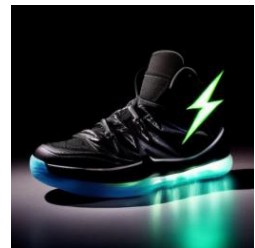

Downtown Seattle at sunrise. detailed ink wash.

A car made out of vegetables.

A sign that says "Diffusion".

A black basketball shoe with a lightning bolt on it.

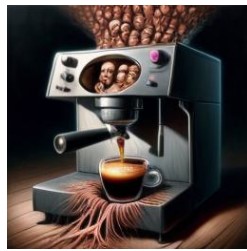
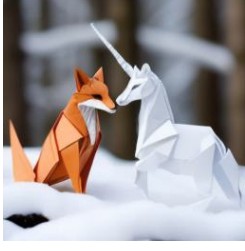
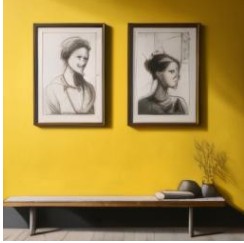
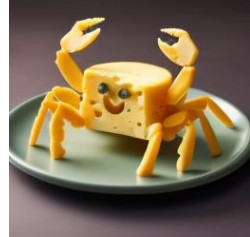

an espresso machine that makes coffee from human souls, high-contrast painting.

Intricate origami of a fox and a unicorn in a snowy forest.

a yellow wall with two framed sketches

A crab made of cheese on a plate.

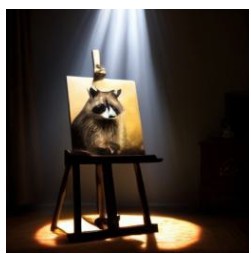
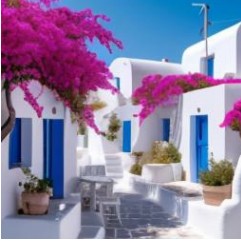
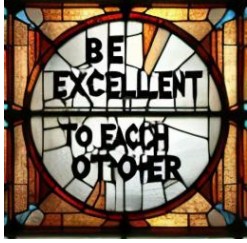
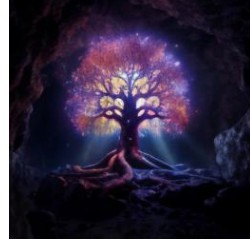

A single beam of light enter the room from the ceiling. The beam of light is illuminating an easel. On the easel there is a Rembrandt painting of a raccoon.

White Cycladic houses with blue accents and vibrant magenta bougainvillea in a serene Greek island setting.

The saying "BE EXCELLENT TO EACH OTHER" written in a stained glass window.

dark high contrast render of a psychedelic tree of life illuminating dust in a mystical cave.

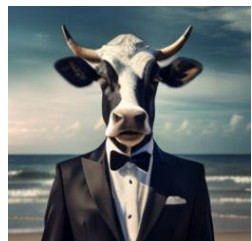
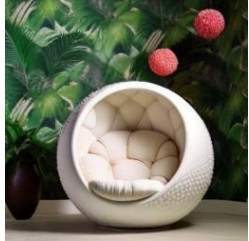
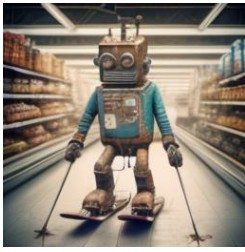
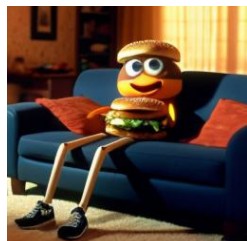

A photo of a person with the head of a cow, wearing a tuxedo and black bowtie. Beach wallpaper in the background.

Photo of a lychee-inspired spherical chair, with a bumpy white exterior and plush interior, set against a tropical wallpaper.

An old rusted robot wearing pants and a jacket riding skis in a supermarket.

Film still of a long-legged cute big-eye anthropomorphic cheeseburger wearing sneakers relaxing on the couch in a sparsely decorated living room.

Figure 8: Generated images from a 7B Transfusion trained on 2T multi-modal tokens.

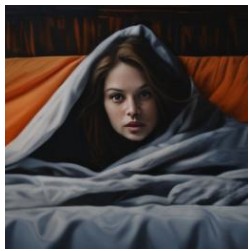
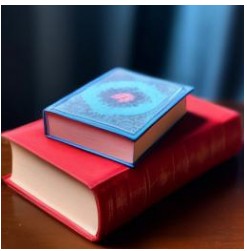
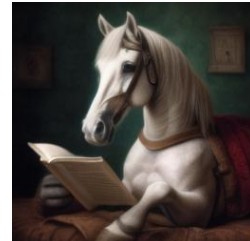
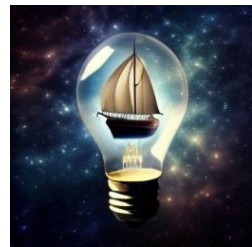

A woman on a bed underneath a blanket.

A small blue book sitting on a large red book.

A horse reading a book.

A light bulb containing a sailboat floats through the galaxy.

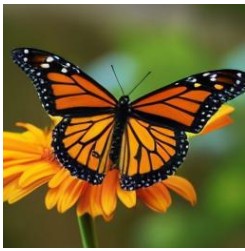
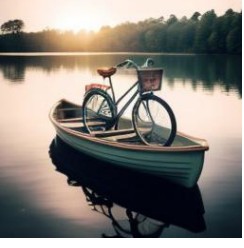
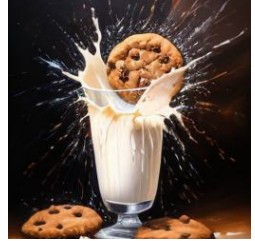
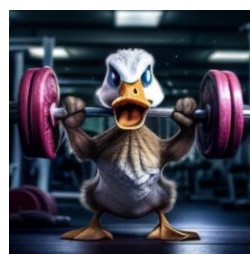

a monarch butterfly.

A rowboat on a lake with a bike on it.

An expressive oil painting of a chocolate chip cookie being dipped in a glass of milk, depicted as an explosion of flavors.

An angry duck doing heavy weightlifting at the gym.

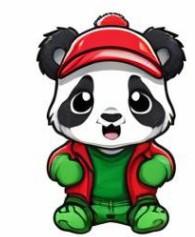
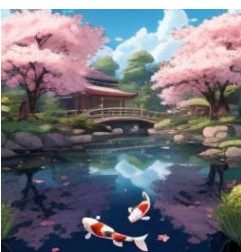
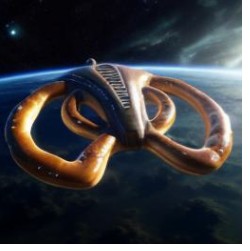
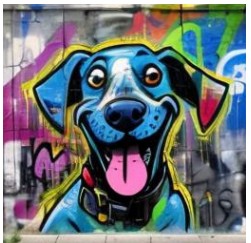

An emoji of a baby panda wearing a red hat, green gloves, red shirt, and green pants.

A tranquil, anime-style koi pond in a serene Japanese garden, featuring blossoming cherry trees.

a massive alien space ship that is shaped like a pretzel.

graffiti of a funny dog on a street wall.

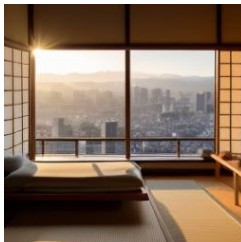
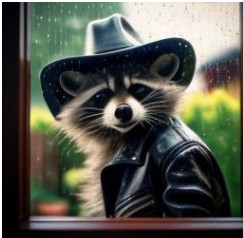
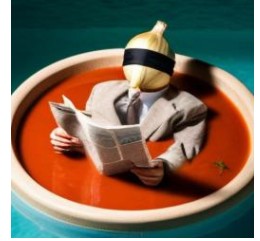
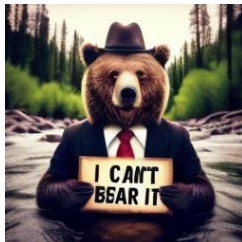

A spacious, serene room influenced by modern Japanese aesthetics with a view of a cityscape outside of the window.

A raccoon wearing cowboy hat and black leather jacket is behind the backyard window. Rain droplets on the window.

A relaxed garlic with a blindfold reading a newspaper while floating in a pool of tomato soup.

photo of a bear wearing a suit and tophat in a river in the middle of a forest holding a sign that says "I cant bear it".

Figure 9: Generated images from a 7B Transfusion trained on 2T multi-modal tokens.

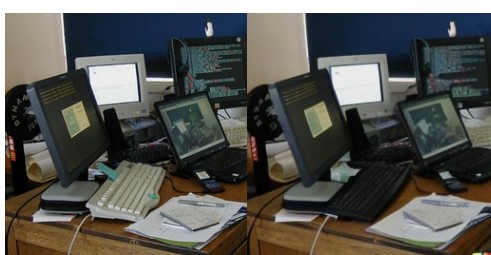

Change the closest keyboard to be all black.

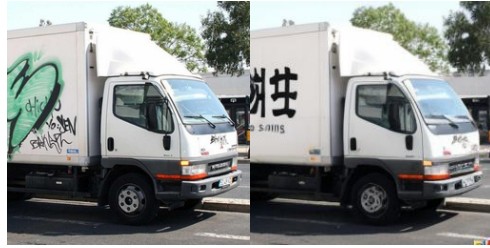

Change the graffiti on the truck into calligraphy writing.

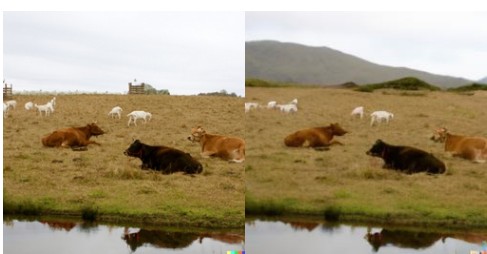

Can we have mountains on the background?

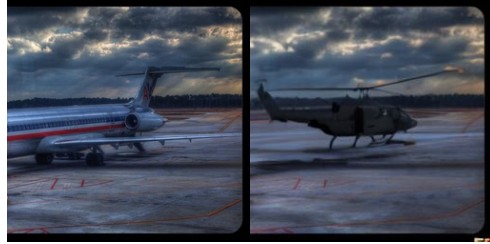

Replace the airplane with a blackhawk helicopter.

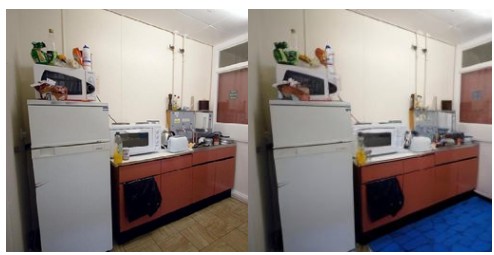

Add a blue rug to the floor.

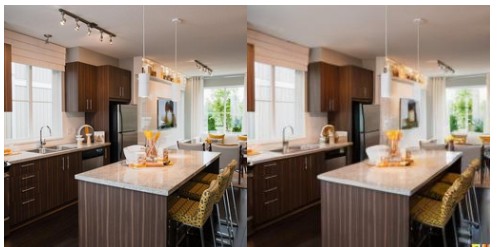

Delete the overhead lights on top of the sink.

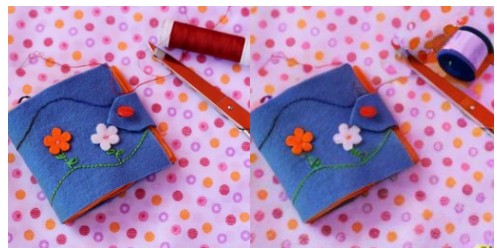

Change the roll of thread into a roll of wire.

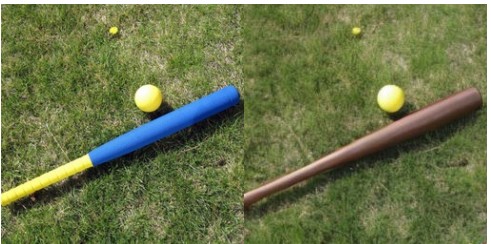

Change the baseball bat to all brown.

Figure 10: Edited images from a fine-tuned 7B Transfusion model.

