# OpenReview forum: "Transfusion: Predict the Next Token and Diffuse Images with One Multi-Modal Model"
_ICLR.cc/2025/Conference — ICLR 2025 Oral_

### Official Review · Reviewer_MTCF · 2024-10-31

**Soundness:** 2
**Presentation:** 3
**Contribution:** 4
**Rating:** 8
**Confidence:** 4

**Summary:**

This paper explores how to bridge the modality gap to inject continuous image generation ability into discrete text-based LLMs by proposing a single transformer model trained with autoregressive modeling and diffusion loss simultaneously. This method of Transfusion avoids image discreteness and still enables input and output-level image tasks. By carefully designing attention masks and input formation, the transformer models text data by AR modeling with causal masking and generates images by denoising with bidirectional attention.

Compared to injecting image comprehension/generation ability by introducing discrete image tokens(Chameleon), the authors demonstrated their approach performs better in various scales in text-to-image generation, image captioning, and even text-only tasks. The large-scale model of Transfusion showed comparable performance to previously published large image generative models.

**Strengths:**

-They suggested a novel training and inference pipeline for a transformer that allows performing both AR and diffusion modeling with shared parameters.

-The manuscript contains extensive controlled experiments and ablation studies.

**Weaknesses:**

-The comparison with previously published works only covered text-only & text-to-image tasks and didn't include their performances on image comprehension tasks(e.g. image captioning).
An additional comparison for these tasks with previous multi-modal LLMs that process images differently(e.g. input-only with an external encoder like LLaVA, or image quantization like Chameleon) would support the effectiveness of Transfusion's approach.

**Questions:**

-How does Transfusion take the embedding of noise timestep in the training/inference of image latent denoising process?

-In the training phase, the model calculates the AR loss and diffusion loss in one forward pass using text tokens and 'noised' image latent, which introduces unavoidable discrepancy when the model inference with clear latent. Wouldn't it be more accurate to calculate AR and diffusion loss separately in two model passes?


-Is there a reason that the COCO FID scores of StableDiffusion models are not in Table 6?

---

> ### Author Response · Authors · 2024-11-27
>
> Q1: In our model, we incorporate the timestep embedding into the denoising process by adding it to each patch vector within the Linear or U-Net-based patch layers. Specifically, before every attention or ResNet block operation, we add the timestep embedding to the corresponding patch representations.
>
> Q2: Thank you for raising this important point. We acknowledge that training with noised image latents introduces a discrepancy during inference when the model processes clear (non-noised) image latents for image understanding tasks. For us to better respond to your question, could you please clarify “calculate AR and diffusion loss separately in two model passes?”. On the other hand, in Table 8 of our paper, we address this issue by limiting the noise level for image understanding tasks, which helps to mitigate the discrepancy. We are conducting additional experiments by eliminating noise for image understanding tasks during training. We will present these ablation results in the revised version of the paper to provide a more comprehensive analysis of this issue.
>
>
> Q3: We appreciate your question. We were unable to include the COCO FID scores for Stable Diffusion models in Table 6 because we could not find the exact FID numbers evaluated on the same dataset under comparable conditions. The official evaluations of Stable Diffusion versions 1.5 and 2.1 available from their GitHub repository are conducted on 10,000 examples, whereas most other works, including ours, report FID scores on 30,000 examples. To ensure a fair and consistent comparison, we refrained from including these results. If we are able to obtain or reproduce the FID scores for Stable Diffusion models on the same 30,000-example COCO dataset, we will include them in the revised paper.

---

> > ### Comment · Reviewer_MTCF · 2024-11-28
> >
> > I thank the clarification from the authors, and would like to add a few more comments:
> >
> > For Q1: Thank you for the clarification, and it would be better to include this explanation in the revised manuscript if it can be done before the revision deadline.
> >
> > For Q2: In the question of “calculating AR and diffusion loss separately in two model passes”, I meant the training phase of "getting diffusion loss with a noise image input, then performing another forward pass with a clear image input to get AR loss" to not introduce training-inference discrepancy at all. This seems to be the ablation study that the authors are currently conducting(eliminating noise for image understanding tasks during training), and it would be appreciated if the authors provide this comparison.

---

> > > ### Author Response · Authors · 2024-11-30
> > >
> > > For Q2, we implemented another version that does the same operations as what you mentioned by "two forward pass". Instead, we duplicate each image twice, the first image is added diffusion noise while the second image is clean. The first image is used to compute the generative diffusion loss while the second image is solely used as a condition for image understanding. We take care of the attention mask, etc, to make sure that the noised image is not seen by future tokens and there is no discrepancies between training and test time. Thus, at test time we can always use clean images for image understanding.
> > >
> > > With this implementation, we can achieve 6 - 8 increase in CiDER scores for image caption task under a controlled setting when comparing to our noised version described in the paper.

---

### Official Review · Reviewer_Huwc · 2024-11-03

**Soundness:** 3
**Presentation:** 4
**Contribution:** 3
**Rating:** 8
**Confidence:** 4

**Summary:**

The paper introduces Transfusion, a novel approach for training multi-modal models that effectively handle both discrete (text) and continuous (image) data. By combining next token prediction for text and diffusion for images within a single transformer architecture, Transfusion eliminates information loss associated with traditional quantization methods. The model was pretrained on a balanced dataset of text and images, utilizing modality-specific encoding and decoding layers to optimize performance. Experimental results demonstrate that Transfusion outperforms existing methods like Chameleon in various benchmarks, achieving significant efficiencies in both text-to-image and image-to-text generation tasks. The study concludes that Transfusion represents a significant advancement in creating integrated multi-modal generative models.

**Strengths:**

1. Innovative Framework: Transfusion introduces a novel approach that seamlessly combines text and image modalities, achieving impressive results in image generation and editing.
2. Robust Comparative Experiments: The paper includes comprehensive comparative analyses and detailed ablation studies that examine various dimensions, such as model architecture and parameters, providing strong evidence for the framework's effectiveness.
3. Clear and Fluid Writing: The writing is clear and the structure of the paper is well-organized, making it easy to follow the authors' arguments and findings.

**Weaknesses:**

Lack of demonstration of multi-modal understanding capabilities: While the paper focuses on image generation and editing capabilities, it does not present extensive benchmarks for the model’s general visual question answering (VQA) abilities. It would be beneficial to include benchmarks related to image understanding, such as VQAV2, SEED Bench and MMMU.

**Questions:**

same as weakness

---

### Official Review · Reviewer_HRoU · 2024-11-04

**Soundness:** 4
**Presentation:** 3
**Contribution:** 3
**Rating:** 8
**Confidence:** 4

**Summary:**

The paper introduces Transfusion, a novel approach for training a single unified model capable of understanding and generating both discrete and continuous modalities. The authors demonstrate that by combining the language modeling loss function (next token prediction) with diffusion, they can train a transformer model over mixed-modality sequences. This method allows for seamless integration of discrete text tokens and continuous image data, without the need for information loss through quantization.

**Strengths:**

1. Originality: The paper introduces a unique approach by combining language modeling and diffusion objectives over shared data and parameters. This method allows for the seamless generation of discrete and continuous modalities.

2. Quality: The paper demonstrates the effectiveness of the Transfusion model through extensive experiments and comparisons with existing methods.

3. Clarity: The paper is well-structured, with clear explanations of the Transfusion method, its architecture, and the training objectives.

4. Significance: The paper addresses a critical challenge in handling both discrete and continuous data modalities.

**Weaknesses:**

#### More experiments are encouraged.
1. Results on multimodal Understanding tasks are limited. For example, VQA, Summarization, VideoQA and etc.
2. I encourage the authors to scale up the model parameters or training data size (Scaling Law on Transfusion is very important), but I understand the computation resources on scaling law are huge. Thus, this is a minor issue.

#### Compare the technical differences of several recent works (e.g., Show-o, MIO, Emu3).

**Questions:**

As shown in Weaknesses.

---

### Official Review · Reviewer_Wjjj · 2024-11-04

**Soundness:** 3
**Presentation:** 3
**Contribution:** 3
**Rating:** 6
**Confidence:** 5

**Summary:**

This paper proposes Transfusion, a unified model for joint understanding and generation. During training, it incorporates CE loss for text learning and diffusion loss for image learning. In inference, it uses casual attention for autoregressive text generation and bidirectional attention for parallel image generation. Compared to Chameleon, Transfusion attains good performance for visual understanding and generation.

**Strengths:**

1. This paper verifies the feasibility combining different learning targets with one framework for respective understanding and generation.

2. It reveals the scaling performance comparing Transfusion and Chameleon, providing good insights for future works.

3. It provides detailed ablation study and analysis to demonstrate the effectiveness of each component.

**Weaknesses:**

1. The visual understanding performance of Transfusion is not convincing enough, as the paper doesn't provide any results on related benchmarks, such as MME, MMBench, etc.

2. The image editing performance also needs quantitative results to show its efficacy.

3. Does this framework has the potential to conduct image-text interleaved generation?

**Questions:**

See weakness

---

### Official Review · Reviewer_BCXk · 2024-11-06

**Soundness:** 3
**Presentation:** 3
**Contribution:** 3
**Rating:** 8
**Confidence:** 3

**Summary:**

The paper proposes a multi-modal transformer model where text tokens are sampled auto regressively and image "tokens" are diffused in a VAE latent space.

**Strengths:**

- problem is well motivated
- method is simple and clear
- results are good
- abalations cover most of my questions

**Weaknesses:**

the method relies on a pre-trained VAE. It would have been nice to see an ablation done in pixel space or with some type of co-training scheme to avoid a 2-stage training process.

**Questions:**

The structure of the UNet is unclear to me. the paper says "since U-Net blocks contain bidirectional attention within, independent of the transformer’s attention mask, this gap is less pronounced when they are applied." What U-Net architecture uses bidirectional attention? Is the UNet applied to each patch independently, or is patching done after the U-Net?

---

> ### Author Response · Authors · 2024-11-27
>
> Thank you for your positive feedback and insightful questions.
>
>
> *Regarding your query about the U-Net architecture and the use of bidirectional attention:*
>
> In our model, we employ a U-Net architecture similar to those used in recent diffusion models (imagen, stable diffusion, dalle2, etc) for image generation. The U-Net consists of downsampling and upsampling blocks, each containing multiple ResNet blocks and attention layers. The attention layers within the U-Net implement bidirectional self-attention mechanisms, allowing every position in the feature map to attend to all others. The CNN layers inside the ResNet are also bidirectional.
>
>
> *Regarding application of the U-Net and Patching:*
>
> The U-Net is applied to the entire image represented in the VAE latent space, not to individual patches. The process is as follows: 1) Downsampling: The input image is processed through the U-Net's downsampling blocks, reducing spatial dimensions while increasing feature channels. 2) Integration with the Transformer: After downsampling, the feature map is reshaped into a sequence of patches (tokens). The transformer applies attention mechanisms over text and image tokens. 3) Upsampling: The transformer's output image tokens are reshaped back into a spatial feature map.This map is processed through the U-Net's upsampling blocks to reconstruct the spatial dimensions, producing VAE latents corresponding to the image.
> In summary, patching occurs after the U-Net's downsampling and before the transformer, and the U-Net operates on the whole image.

---

### Author Response · Authors · 2024-11-27
**General Response**

*Response to General Concern: Lack of Demonstration of Multimodal Understanding Capabilities*

We appreciate the reviewers' valuable feedback and acknowledge the concern regarding the absence of benchmarks demonstrating Transfusion's multimodal understanding capabilities, such as VQAv2, SEED-Bench, and MMEU. Our primary focus in this paper was to compare different multimodal pretraining methods and investigate their scaling laws.

In our study, we highlighted image editing tasks because they represent scenarios not covered during pretraining. We aimed to explore whether Transfusion could learn such capabilities through fine-tuning. Due to limited resources, we were unable to extensively explore visual instruction tuning or include comprehensive evaluations on image understanding benchmarks.

In future work, we plan to curate a robust set of visual instruction data, explore model extensions (such as integrating pretrained image encoders) and effective training recipes. By addressing these areas, we aim to provide a more thorough evaluation of Transfusion's performance on established benchmarks in our subsequent work. We believe this will demonstrate the model's full potential in multimodal understanding tasks and appreciate your suggestion to include these evaluations.

---

### Meta-Review · Area_Chair_KCef · 2024-12-05

**Metareview:**

**Summary**. The paper introduces Transfusion, a unified multi-modal model trained jointly on text and image data. By combining next-token prediction for text and diffusion-based learning for images within a single transformer architecture, it bridges the modality gap without quantizing images into discrete tokens. The study establishes scaling laws for this architecture and evaluates its performance against other multi-modal models such as Chameleon. The authors demonstrate that Transfusion achieves state-of-the-art results in text-to-image generation, while maintaining competitive performance on text-only tasks.

**Strengths**.
(1) Innovative Idea: The combination of autoregressive modeling for text and diffusion processes for images within one model is novel and effectively addresses the modality gap.
(2) Comprehensive Experiments: Extensive ablation studies and comparisons with state-of-the-art models provide strong evidence for the validity and robustness of the proposed method.
(3) Scaling Insights: The paper provides insights into scaling laws, which are valuable for guiding future research in multi-modal modeling.
Clear Presentation: The manuscript is well-structured, with detailed explanations of methods and results, making it accessible to a broad audience.

**Weaknesses**.
(1) Limited Multi-Modal Understanding Results: The paper lacks comprehensive benchmarks on tasks like visual question answering (e.g., VQAv2, MMEU), which are critical for assessing multi-modal understanding.
(2) Image Editing Evaluation: Quantitative results for image editing tasks are not sufficiently detailed, limiting the assessment of this capability.
(3) Training Discrepancies: The reliance on noised image latents during training for image understanding tasks introduces discrepancies with inference, though this was partially addressed in the rebuttal. Comparisons with Recent Methods: Technical differences with some contemporary works (e.g., Show-o, MIO, Emu3) are not discussed thoroughly.

**Reasons**
The paper presents a very novel and effective method for multi-modal modeling for joint understanding and generation, supported by extensive experiments and insights into scaling behavior. Despite some limitations in demonstrating comprehensive multi-modal understanding and minor training discrepancies, the strengths far outweigh the weaknesses. The clarity of presentation and significance of contributions to the field justify its acceptance.

**Additional Comments On Reviewer Discussion:**

The reviewers raised several pertinent points, particularly regarding the lack of benchmarks for multi-modal understanding and the discrepancy in training versus inference for image understanding tasks. The authors addressed these concerns in their rebuttal:

- Benchmarks for Multi-Modal Understanding: The authors acknowledged this gap and proposed to include more benchmarks in future work, focusing on visual instruction tuning.
- Training Discrepancies: The authors introduced an alternative training method involving dual forward passes (clean and noised images), achieving significant performance improvements for image understanding tasks like captioning.
- U-Net Architecture and Patch Processing: Clarifications on the U-Net's bidirectional attention and patching process provided a deeper understanding of the model architecture.

Overall, the discussion demonstrated the authors' responsiveness and willingness to acknowledge and address weaknesses (to some extent), improving confidence in the paper's quality.

---

### Decision · Program_Chairs · 2025-01-22

Accept (Oral)